# Psychometric characteristics and factorial structures of the Defensive Pessimism Questionnaire—Spanish Version (DPQ-SV)

Carmen Ramírez-Maestre[1]*, Rosa Esteve[1], Elena R. Serrano-Ibáñez[1], Alicia E. López-Martínez[1], Gema T. Ruiz-Párraga[1], Teresa Rivas-Moya[2]

1 Facultad de Psicología y Logopedia, Andalucía Tech, Instituto de Investigación Biomédica de Málaga (IBIMA), Universidad de Málaga, Málaga, Spain, 2 Departamento de Psicobiología y Metodología de las Ciencias del Comportamiento, Facultad de Psicología, Universidad de Málaga, Málaga, Spain

* cramirez@uma.es

**Data Availability Statement:** All relevant data are available at https://data.mendeley.com/datasets/

## Abstract

The aim of this study was to validate the Spanish version of the Defensive Pessimism Questionnaire. A sample of undergraduate students (N = 539) was measured on defensive pessimism using the Defensive Pessimism Questionnaire (DPQ), optimism and pessimism using the Life Orientation Test (LOT), positive and negative affect using the Positive and Negative Affect Schedule, and anxiety using the trait subscale of the State and Trait Anxiety Inventory. A Spanish version of the DPQ (DPQ-SV) is presented. Exploratory and Robust Confirmatory Factor Analysis had a bi-dimensional structure (Reflectivity and Negative Expectation). Omega coefficient showed a high internal consistency and the temporal stability was high in each dimension. Both DPQ-SV subscales (Negative Expectation and Reflectivity) showed adequate convergence with LOT-optimism and LOT-pessimism. Reflectivity showed adequate criterion validity with trait-anxiety and negative affect, but inadequate criterion validity with positive affect. Negative Expectation showed excellent criterion validity with trait-anxiety and negative affect and good criterion validity with positive affect. Finally, mediation analysis showed that Negative Expectation had a significant indirect mediating effect between trait-anxiety and negative affect. Reflectivity had a significant indirect mediating effect between trait-anxiety and negative and positive affect. Analysis of the psychometric properties of the DPQ-SV subscale scores showed that it is a two factor adequate measurement tool for its use in this type of samples.

## Introduction

Dispositional optimism has been described as a bipolar dimension anchored by optimism (i.e., the tendency to believe that one will generally experience good versus bad outcomes in life) and pessimism (i.e., the expectation of negative outcomes) [1]. However, it has been argued that pessimism and optimism are independent constructs that are moderately to strongly correlated and that should be treated separately [2–3].

zvj6cz68sc/1 or http://dx.doi.org/10.17632/
zvj6cz68sc.1.

**Funding:** This research was supported by grants
from from the Spanish Ministry of Science and
Innovation (PSI2013-42512- P), and the Regional
Government of Andalusia (HUM-566; CTS-278).
The funders had no role in study design, data
collection and analysis, decision to publish, or
preparation of the manuscript.

**Competing interests:** The authors have declared
that no competing interests exist.

In addition, two types of optimism and pessimism have been distinguished: defensive pessi-mism and strategic optimism [4–5]. Individuals with high scores in defensive pessimism set unrealistically low expectations for upcoming performance, even though they have had previous successes, and then devote considerable energy to mentally playing through or reflecting on all the possible outcomes they can imagine for a given situation. Thus, defensive pessimism includes two different domains: negative expectations about success in a particular task (pessi-mism) and reflection [6]. These two variables may act independently in the behavior of defensive pessimists [7]. The role of reflection is complex and has two different explanations: the *dissipa-tion hypothesis* and the *harnessing hypothesis* [8]. The *dissipation hypothesis* suggests that when defensive pessimists reflect about worst-case scenarios, they dissipate negative affect before per-formance. However, according to the *harnessing hypothesis*, when they reflect about potential negative outcomes regarding an upcoming task there is an increase in negative affect (anxiety), which helps defensive pessimists to focus on developing good performance [7; 9]. More empiri-cal research is needed to support one of these two theoretical explanations. Nevertheless, the role of anxiety is crucial in the defensive pessimism process. Defensive pessimism could be a cogni-tive strategy with two domains that anxious individuals use in the face of challenging situations [4,10]. Gasper et al. [7] found that anxiety was a central variable in the defensive pessimistic pro-cess. Therefore, anxiety could both influence and follow from their goal appraisals. Furthermore, defensive pessimism could be considered a strategy used by individuals concerned about failure and focused on achieving a good performance [10]. According to Ferradás, et al. [11] defensive pessimism is common among students with a high degree of academic self-demand.

In the 1980s, Norem and Cantor [5] developed a unidimensional questionnaire (negative expectations) to assess defensive pessimism. Subsequently, the Defensive Pessimism Question-naire (DPQ) included two correlated factors called *Reflectivity* and *Pessimism* that assessed both domains: negative expectations and reflection [8]. Although the DPQ has been viewed as a unidimensional measure of defensive pessimism [12], factorial analysis shows that negative expectations and reflection are different but correlated factors [13].

The psychometric properties of the DPQ have not been tested in a Spanish sample. Thus, the aim of this study was to analyse the psychometric properties (reliability and validity) of the Spanish version of the Defensive Pessimism Questionnaire (DPQ-SV) in a sample which com-prised 539 undergraduate students. The results could deepen our understanding of defensive pessimism and of the scale itself. The factor structure, internal consistency, convergent validity and criterion validity of the DPQ-SV were evaluated. Convergent validity was assessed between the DPQ-SV subscales scores and optimism and pessimism scores on the Life Orien-tation Test (LOT). According to theoretical considerations [4, 10], a moderate-high association was expected between negative expectation scores, LOT-pessimism scores (positive) and LOT-optimism scores (negative), and a low-moderate correlation between reflectivity scores, LOT-pessimism scores (positive) and LOT-optimism scores (negative). In order to test concurrent criterion validity, we measured the association between defensive pessimism and affect. According to the harnessing hypothesis [8], negative expectations and reflection increase nega-tive affect, which helps defensive pessimists to obtain better performance outcomes. Therefore, we expected to find moderate-strong positive correlations between DPQ-SV subscale scores and negative affect, and moderate-strong negative correlations between DPQ-SV subscale scores and positive affect. Finally, it has been suggested that defensive pessimists are typically high in trait-anxiety [4, 5, 14]. It has also been found that defensive pessimism, understood as a cognitive strategy that anxious individuals use in the face of challenging situations, increases negative affect and decreases positive affect [4,6,10]. Therefore, in order to test the validity of the DPQ-SV, we also assessed the mediating role of the dimensions of defensive pessimism between trait-anxiety and affect.

## Materials and methods

### Participants

During the course 2017/18, random cluster sampling was conducted among the 1080 undergraduate students of the Faculty of Psychology of Malaga University. Firstly, two groups from each course were randomly selected. The questionnaires were administered to all students in each group. The sample size was 551 students from the four courses (199, 139, 113, 100 students from each course, respectively). Twelve participants had missing scores and so the final sample size comprised 539 students. The inclusion criterion was the ability to understand the Spanish language. Women comprised 78.8% of the sample. Mean age was 21.18 years (SD = 4.53; range 18–53)

All participants were fully informed of the aim of the study, and given guarantees of personal anonymity and the confidentiality of the survey. Subsequently, their consent was obtained to voluntarily participate in the study. Part of the sample are the same that the ones included in a previous research [15].

### Procedure

Three psychologists took part in data collection. They were trained in the application of the protocol to guarantee the standardization of the assessment process. The students were always assessed in their usual classroom. Teachers and students were informed of the aim of the research before they agreed to participate. The survey was voluntary and students who completed it does not received course credits for their participation. The consent form was explained on the first screen of the survey and all participants were required to agree to participate in the study before they continued and completed the research protocol.

The Ethics Committee of the University of Málaga approved this study (CEUMA 2013-0016-H).

### Measures

**Defensive Pessimism Questionnaire (DPQ).**   The published Spanish translation of the Defensive Pessimism Questionnaire [6] was used. This instrument contains 12 items (e.g., "Considering everything that can go wrong helps prepare me"). Participants respond on a Likert scale ranging from 1 (*not at all true of me)* to 7 (*very true of me)*.

**Life Orientation Test-revised.**   The Spanish version [16] of the Life Orientation Test-Revised (LOT-R) [1] was applied to measure dispositional optimism and pessimism. The LOT-R comprises 10 items: 4 filler items, 3 measuring optimism (e.g., "Overall, I expect more good things to happen to me than bad"), and 3 measuring pessimism (e.g., "I hardly ever expect things to go my way"). Participants respond on a Likert scale ranging from 0 (*strongly disagree*) to 4 (*strongly agree*). Optimism and pessimism items were summed such that higher scores indicated higher levels of optimism and pessimism, respectively. In this study, the omega values were .75 for optimism and.68 for pessimism.

**The positive and negative affect schedule.**   The Positive and Negative Affect Schedule (PANAS) [17], adapted to Spanish by Sandín et al. [18], was administered to measure the extent to which individuals normally feel a range (frequency) of positive and negative affect. This scale comprises 20 items: 10 for each of the two dimensions. Participants respond on a 5-point Likert-type scale ranging from 1 (*never*) to 5 (*always*). Positive and negative affect items were summed such that higher scores indicated higher levels of positive and negative affect, respectively. In this study, omega values for positive and negative affect were.89 and .86, respectively.

**State and Trait Anxiety Inventory (STAI).** The STAI [19] comprises two 20-item scales that assess anxiety as a trait and anxiety as a state. Only the STAI-Trait subscale was used in this study. The STAI-Trait addresses how respondents "generally feel" (e.g., "I am a steady person"; "I lack self-confidence"). Respondents are asked to rate themselves on each item on a 4-point Likert scale, ranging from *almost never* to *almost always*. The Spanish version of the STAI has excellent construct and criterion validity [20]. In the present study, the STAI-Trait scale had an omega coefficient equal to .91.

## Data analyses

The number of dimensions was assessed using Indices based on Parallel Analysis (PA), Very Simple Structure (VSS), and Minimum Average Partial Correlation (Velicer's MAP). The dimensionality of the DPQ-SV items was conduced using Exploratory Factor Analysis (EFA) —principal axis method–and Robust Confirmatory Factor Analysis (RCFA) using a robust weighted least squares (WLSMV) estimation method, indicated for categorically ordered data, and a polychoric correlation matrix.

Goodness of fit was evaluated using the following indices [21]: standardized root mean square residual (SRMR), root mean square error of approximation (RMSEA), the comparative fit index (CFI), the Tucker-Lewis index (TLI) and the weighted root mean residual (WRMR). Model fit was defined by the following criteria: RMSEA value equal to .05 or less is considered a good fit, .08 for acceptable fit, and .10 or more a poor fit [22]. SRMR value close to .08 or below for acceptable fit [23], and CFI [24] and TLI values should be greater than or close to .95. Yu and Muthén [25], recommend the WRMR over the SRMR for categorical indicators, with good fit at values close to 1.00 and below.

The reliability and analysis of the items were assessed using the omega coefficient (ω) and 'ω if each item is deleted'. The 95% Confidence intervals (CIs) for ω and ω (-item) were calculated. Test-retest reliability was analysed using Pearson correlation coefficients.

Convergent validity of the DPQ-SV sub-scales was assessed by computing the Pearson-correlation coefficients between the DPQ-SV dimensions and optimism and pessimism LOT scores.

Concurrent criterion validity was assessed by calculating Pearson correlations between the DPQ-SV dimensions and the negative and positive PANAS scores.

These analyses were performed using different R packages: *psych* version 1.7.3.21 [26], *paran* version 1.5.1 [27], *MBESS* version 4.2.0 [28], *lavaan* 0.5–12 [29].

In addition, the procedure described by Preacher and Hayes [30] was applied to investigate the role of defensive pessimism as a mediator between trait-anxiety and affect (positive and negative). The mean direct and indirect effects and their CIs were calculated using the estimates based on 1000 bootstrap samples.

The mediation analysis was conducted using the SPSS Statistic package version 22.0.

## Results

### Preliminary analysis

Descriptive statistics for items can be seen in Table 1. Using an cutoff of ±1 for skew and 0 for kurtosis, all items had skewness <±1 except for item 2 (1) and item 10 (-1.05) . All items had kurtosis greater or lower than cero, but there is no item with highly leptokurtic distribution. Item 2 and item 10 had each one 10 outliers (1.5xIQR rule). Kolmogorov-Smirnov's test shows that none of items followed univariate normality.

**Table 1. Item analysis and reliability of the DPQ-SV (N = 539).**

| Item | M | SD | Skew | Kurtosis | $\omega$ (-item) | $CI\ \omega$ (-item) |
|---|---|---|---|---|---|---|
| r3 | 5.26 | 1.39 | -.72 | .19 | .79 | .76-.82 |
| r7 | 4.78 | 1.64 | -.61 | -.38 | .75 | .71-.79 |
| r8 | 4.50 | 1.65 | -.40 | -.70 | .77 | .73-.80 |
| r9 | 4.42 | 1.82 | -.34 | -.96 | .76 | .72-.79 |
| r10 | 5.32 | 1.57 | -1.05 | .38 | - | - |
| r12 | 4.47 | 1.72 | -.35 | -.86 | .78 | .75-.81 |
| Reflectivity | | | | | $\omega$ = .81 | $CI\ \omega$:.78-.84 |
| e1 | 3.71 | 1.93 | -.06 | -1.18 | .85 | .82-.87 |
| e2 | 5.41 | 1.40 | 1.00 | .77 | .85 | .83-.87 |
| e4 | 4.53 | 1.87 | -.33 | -1.10 | .83 | .80-.85 |
| e5 | 3.59 | 1.94 | .28 | -1.21 | .80 | .78-.83 |
| e6 | 3.75 | 1.95 | .11 | -1.24 | .83 | .80-.85 |
| e11 | 3.46 | 1.93 | .25 | -1.60 | .87 | .84-.88 |
| Negative Expectation | | | | | $\omega$ = .86 | $CI\ \omega$:..84-.88 |

Mean (*M*), Standard Deviation (SD), Confidence Interval 95% (*CI ω* (-item)), Confidence Interval 95% (*CI ω*)

## Exploratory Factor Analysis

The Kaiser-Meyer-Olking (KMO) index was .89. The subject-to-item ratio was 45:1, indicating that EFA was adequate for this sample.

Several indices suggested one factor (Parallel Analysis, VSS Complexity 1, being Maximum = .96) or two factors (VSS Complexity2, being Maximum = .75, Velicer´s MAP, being Minimum = .03).

Table 2 shows the EFA results for Model 1 (one factor with item 10), Model 2 (one factor without item 10), Model 3 (two factors with item 10) and Model 4 (two factors without item 10). The Model 1 solution accounted for 38.69% of the variance. All loadings were greater than

**Table 2. Exploratory Factor Analysis.**

| Item | Model 1 | | Model 2 | | Model 3 | | | Model 4 | | |
|---|---|---|---|---|---|---|---|---|---|---|
| | Load | $h^2$ | Load | $h^2$ | Load1 | Load2 | $h^2$ | Load1 | Load2 | $h^2$ |
| e1 | .62 | .38 | .62 | .39 | **.69** | .03 | .45 | **.68** | -.03 | .42 |
| e2 | .63 | .39 | .62 | .39 | **.45** | .25 | .38 | **.42** | .24 | .40 |
| r3 | .44 | .19 | .43 | .19 | -.05 | **.65** | .39 | -.21 | **.74** | .31 |
| e4 | .73 | .53 | .73 | .54 | **.77** | .01 | .61 | **.80** | -.02 | .55 |
| e5 | .80 | .64 | .81 | .65 | **.90** | -.04 | .78 | **.91** | -.05 | .71 |
| e6 | .76 | .57 | .76 | .57 | **.72** | .10 | .60 | **.70** | .10 | .63 |
| r7 | .63 | .40 | .63 | .39 | .18 | **.62** | .53 | .00 | **.74** | .45 |
| r8 | .69 | .48 | .69 | .48 | .42 | **.38** | .48 | .30 | **.46** | .44 |
| r9 | .67 | .45 | .67 | .45 | .26 | **.57** | .54 | .13 | **.63** | .47 |
| r10 | .22 | .05 | | | -.08 | **.38** | .12 | | | |
| e11 | .49 | .24 | .49 | .24 | **.56** | -.05 | .29 | **.60** | -.09 | .28 |
| r12 | .58 | .33 | .57 | .33 | .23 | **.48** | .39 | .09 | **.56** | .35 |
| %Var | 38.69 | | 41.79 | | 39.29 | 6.88 | | 39.00 | 34.00 | |

Communality($h^2$)

.40 and communalities were between .19 and .64, except for item 10 (loading .22; communality .05).

If item 10 is deleted, the one-factor solution (Model 2) accounted for 41.79% of the variance, and all loadings were greater than .49.

The Model 3 solution (promax rotation) accounted for 39.29% and 6.88% of the variance and correlation between factors was .67. Item 8, being a Reflexivity item, had a loading (.42) on the Negative Expectation factor and a loading (.38) on the Reflectivity factor. Item 10 had a loading (.38) lower than the threshold .40 on the Reflectivity factor.

In the Model 4 two-factor solution (promax rotation) each factor accounted for 39% and 34% of the variance, and correlation between factors was .68. Items 1, 2, 4, 5, 6, and 11 define the Negative Expectation dimension and items 3, 7, 8, 9, and 12 define the Reflectivity dimension. All loadings were greater than the threshold .40.

Findings from EFA supported a one-factor structure (Model 2) and a two-factor structure (Model 4). Two-factor structure explaining a greater variance (73%) of the items than one-factor solution (41.79%).

## Robust Confirmatory Factor Analysis

All polychoric correlations between items were below .65 except for a high correlation (r = .83) between r5 and r6 items, then that there was no multicollinearity in the dataset was concluded. This high correlation could be due to some content overlap between the two items.

Mardia's multivariate tests showed that none of the latent variables were multivariate normal (S1 Table) and a visual inspection of the data (S1–S5 Figs) suggested the presence of multivariate outliers and the violation of multivariate normality in all latent variables.

Models 1–4 were tested with RCFA and the method above indicated. Table 3 shows fit indices in the four models. The tests based on $\chi^2$ were significant (p < .001) (Table 3, Column2), but these values are sensitive to sample size, non-normality and large correlations between variables [31]. Bold indices (SRMR, WRMR, CFI, TLI) indicate acceptable fit. Model 3 and Model 4 (Table 3, rows 3–4) shows better fit indices than Model 1 and Model 2 (Table 3, rows 1–2). In Model 3 and Model 4, there is an acceptable goodness of fit. In Model 4, TLI (.96), CFI (.97), SRMR (.051) and WRMR (1.10) values all approximated good fit according to the criteria described above. The RMSEA value (.097) indicated an acceptable model fit (< .10;[17]) and surpassed Hu and Bentler's [23] criteria for good model fit. Chen et al. [32] recommended that RMSEA values be considered in the context of other fit indices rather than solely in terms of universal cutoff points. On this basis, and considering that the other four fit indices consistently indicated a good fit, that the Model 4 had acceptable fit was concluded. Then, Model 4 shows significantly/ slight better fit indices than Model 3.

Fig 1 shows the path diagram of the Robust Confirmatory Factor Analysis for Model 4. Item standardized loadings (>.40) on each of the two factors, estimated variances of the items were >.40 except for e4 (.38), e5 (.15) and e6 (.29), and a high correlation (.77) between Reflectivity and Negative Expectation factors are shown.

**Table 3. Robust Confirmatory Factor Analysis fit indices.**

| Model | $\chi^2$ (df) | RMSEA | SRMR | WRMR | CFI | TLI |
|---|---|---|---|---|---|---|
| Model 1 | 640.93(54) | .142 | .078 | 1.71 | .91 | .90 |
| Model 2 | 583.48(44) | .151 | .075 | 1.69 | .91 | .89 |
| Model 3 | 312.65(53) | **.095** | **.056** | **1.17** | **.96** | **.95** |
| Model 4 | 262.51(43) | **.097** | **.051** | **1.10** | **.97** | **.96** |

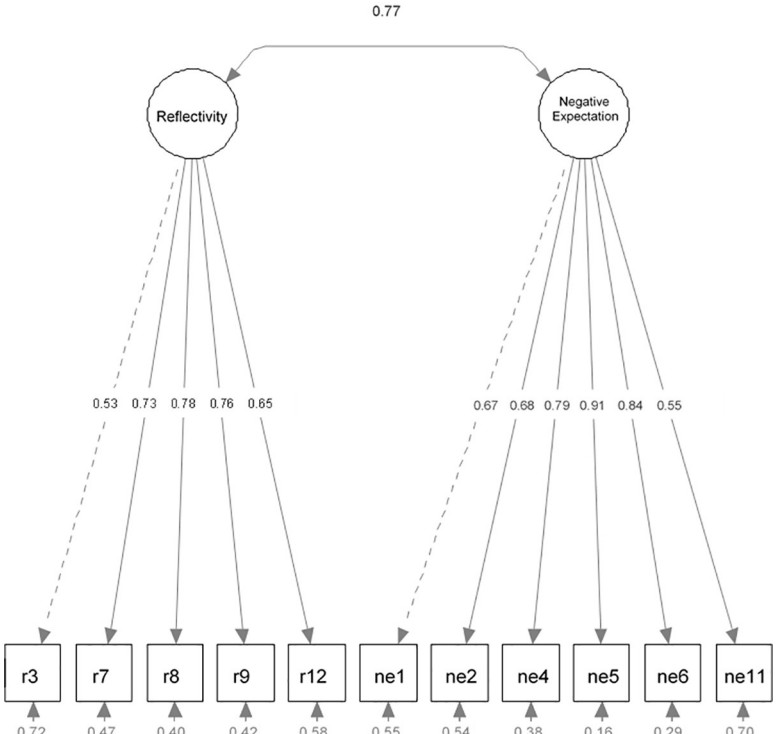

**Fig 1. Path diagram of the Robust Confirmatory Factor Analysis.**

## Reliability

In Model 4 solution, Internal Consistency of the scale was indicated by a $\omega$ coefficient of .81 (95% CI, .78-.84) for the Reflectivity scale, and .86 (95% CI, .84-.88) for the Negative Expectation scale. Internal Consistency showed satisfactory values of reliability in the sample and the 95% CI values satisfactory values of reliability in the population.

Indices for each item—' $\omega$ if an item was deleted'- and 95% CI for '$\omega$ if an item was deleted'—are shown in Table 1. These $\omega$ coefficients of each scale did decrease, when an item was deleted, regarding to $\omega$ of scale, then all the items contributed to the Internal Consistency in each scale.

## Test-retest reliability

Temporal stability was assessed in a subgroup (N = 84) of the sample. This subgroup answered the DPQ-SV twice, with approximately 1 month between sessions. A correlation coefficient ($r$ = .73 CI [.61, .81]; p < .001) was found for the association between the Reflectivity scores at Time-1 (*M* = 25.10, *SD* = 6.00, *Range* 10–35) and at Time-2 (*M* = 24.71, *SD* = 5.52, *Range* 12–34). A correlation coefficient ($r$ = .82 CI [.74, .88]; p < .001) was found for the association between the Negative Expectation scores at Time-1 (*M* = 26.26, *SD* = 8.39, *Range* 8–42) and at Time-2 (*M* = 27.35, *SD* = 8.35, *Range* 11–42). In both factors correlations were high and significant, suggesting adequate temporal stability.

## Other evidence of construct validity

We analysed other evidence of construct validity of the two components of defensive pessimism.

Convergent validity was assessed by computing the Pearson correlation coefficients between the DPQ-SV sub-scale scores and pessimism and optimism scores of the LOT sub-scales (LOT-pessimism, *M* = 4.9, *SD* = 2.4, *Range* 0–12, N = 538; LOT-optimism, *M* = 7.5, *SD* = 2.4, *Range* 0–12, N = 539).

According to Evers et al. [33], when a correlation has been found between two very similar instruments, a convergent validity value can be considered inadequate (r < .55), adequate (.55 ≤ r < .65), good (.65 ≤ r < .75), or excellent (r ≥ .75). However, when the instruments (constructs measured) are less similar, lower values may be adequate. Thus, a negative high association (r = -.56 CI [-.62, -.50]; p < .001) was found between the Negative Expectation and LOT-optimism subscale scores and a positive high correlation (r = .61 CI [-.62, -.50], p < .001) was between the Negative Expectation and LOT-pessimism subscale scores. As expected, a negative low association was also found between Reflectivity and LOT-optimism subscale scores (r = -.23 CI[-.31, -.14]; p < .001), and a positive low association between Reflectivity and LOT-pessimism subscale scores (r = .33 CI[-.40, -.25]; p < .001). All these correlations reached statistical significance (*P* < .001).

Therefore, both DPQ-SV subscales (Negative Expectation and Reflectivity) showed adequate convergence with LOT-optimism and LOT-pessimism. In addition, the two components of defensive pessimism explained a higher average variance percentage of the LOT-pessimism scores (24%) than that of the LOT-optimism scores (18.33%).

## Criterion validity

Concurrent criterion validity was assessed by computing Pearson correlations between the DPQ-SV subscales and Trait-anxiety, (STAI-trait, *M* = 23.2, *SD* = 10.8, Range: 3–56, N = 531) and the Negative affect scores (PANAS-negative, *M* = 21.7, *SD* = 6.7, Range: 10–50, N = 538).

According to Evers et al. [33], a criterion validity value can be considered inadequate (r < .20), adequate (.20 ≤ r < .35), good (.35 ≤ r < .50), or excellent (r ≥ .50). A positive moderate association (r = .36 CI [.28, .43]; p < .001) was found between the Reflectivity and Trait-anxiety subscale scores and a positive high association between the Negative Expectation scores and Trait-anxiety subscale scores (r = .72 CI [.68, .76]; p < .001). A negative very low and non-significant association (r = -.05 CI [-.13, .03]; p = .24) was found between the Reflectivity and PANAS-positive subscale scores and there was a positive moderate correlation (r = .34 CI [.26, .41]; p < .001) between the Reflectivity and PANAS-negative subscale scores. A negative moderate association was found between the Negative Expectation, and PANAS-positive subscale scores (r = -.45 CI [-.51, -.37]; p < .001) and a positive high correlation was between both Negative Expectation and PANAS-negative subscale scores (r = .60 CI [.54, .65]; p < .001).

Reflectivity and Negative Expectation scores showed good or excellent criterion validity with Trait-anxiety scores, respectively. Reflectivity scores showed adequate criterion validity with PANAS-negative scores but inadequate criterion validity with PANAS-positive scores. Negative Expectation scores showed excellent criterion validity with PANAS-negative scores and good criterion validity with PANAS-positive scores, respectively.

## Mediation analysis

Table 4 summarizes the results of the Multiple Mediator Analysis, and shows the path coefficients and confidence intervals for each effect tested in the model. It was analysed the indirect mediating association of the two components of defensive pessimism. The results of the Multiple Mediator Analysis showed that Negative Expectation had a significant indirect mediating association between trait-anxiety and negative affect. However, the indirect mediating association of Negative Expectation between trait-anxiety and positive affect was non-significant.

**Table 4. Path coefficients and confidence intervals of mediational analyses.**

| Independent variable (IV) | Mediating variable (M) | Dependent variable (DV) | Effect of IV on M | Total effect | Direct effect | Indirect effect | 95% CI for indirect effect |
|---|---|---|---|---|---|---|---|
| Anxiety | Negative expectation | PA | .56** | .42**.4 | .41**. | -.003 | -.05 to |
| | | NA | | 4** | 37** | ns.07** | .04.02 to .11 |
| Anxiety | Reflectivity | NA | .20** | .44** | .42** | .02** | .00 to .03 |

Estimated using bias corrected and accelerated bootstrapping, with 5.000 samples.

CI = confidence interval. PA = Positive affect; NA = Negative affect.

** $p < .001$; ns = non-significant

Reflectivity showed a significant mediating association between trait-anxiety and negative affect. Since the correlation between reflectivity (mediator) and positive affect (criterion variable) was nonsignificant, according to Baron and Kenny [34], the mediator role of reflectivity between trait-anxiety and positive affect could not be calculated.

## Discussion

The aim of this study was to provide empirical evidence regarding the psychometric qualities of the DPQ-SV. The results are in line with those described in the study by Lim [13], who found that the most appropriate solution would be a two factor solution without item 10 ("I imagine how I would feel if things went well"). Lim [13] suggested that item 10 may not adequately capture the essence of the defensive pessimism construct, given that defensive pessimists are concerned about the possibility of negative events but do not consider positive outcomes. However, Norem [8] argued that people who tend to reflect extensively about possible negative outcomes also tend to reflect extensively about possible positive outcomes, and that both thoughts are part of the reflection process of defensive pessimists. In any case, according to our results, the quality of the DPQ could be increased without item 10. Our results, and those of Lim [13], could simply be the consequence of the existence of only one item assessing reflection about positive outcomes; if more items regarding positive outcomes had been included, the results of the factor analysis may not have indicated the exclusion of item 10.

Previous research has shown that individuals high in pessimism (both defensive and typical) are characterized by negative expectations about the future: however, whereas typical pessimists are passive, defensive pessimists reflect and plan how to manage future events [8]. The union of both Negative Expectations and Reflectivity differentiate defensive and typical pessimism. Thus, although the DPQ has been viewed as a unidimensional measure of defensive pessimism [12], factorial analysis has found two correlated factors [6, 8, 13]. Our results support this finding. Norem & Chang [8] suggested more differences between defensive pessimism and pessimism as conceived by Carver et al [2]. Norem stated that defensive pessimism is a strategy (relatively malleable) used to prepare for stressful events, whereas Carver et al. [2] conceived of pessimism as a stable trait. In this study, correlation analyses showed a significant positive correlation between the DPQ-SV factors and LOT-pessimism.

Individuals who use defensive pessimism typically have high levels of anxiety [4]. In the present study, there are high-moderate correlations between both defensive pessimism factors and trait-anxiety. We also found that negative expectations and reflectivity factors mediated between trait-anxiety and negative affect. These results support those of Araújo et al. [35], who analysed the relationship between defensive pessimism and psychological health in a sample of 192 university students. These authors found that the low expectations set by defensive

pessimists fully mediated the relationship between anxiety and poor psychological health. On the other hand, we have found that both Negative Expectations and Reflectivity had a positive (high and moderate respectively) correlation with negative affect. Interestingly, the correlation between reflectivity and positive affect was not significant. In line with previous studies, these results support the *harnessing hypothesis* versus the *dissipation hypothesis* [7, 9]. According to the *dissipation hypothesis*, when defensive pessimists reflect about worst-case scenarios, they dissipate negative affects before performance. In relation to this theoretical framework, Norem & Chang [8] suggested that individuals with high levels of anxiety could use defensive pessimism as a way to cope with their negative affect to achieve successful performance. Norem & Cantor [4] also proposed that defensive pessimists control their anxiety by thinking through possible outcomes prior to an event. However, according to the *harnessing hypothesis*, when defensive pessimists reflect on potential negative outcomes for an upcoming task there is an increase in negative affect, which helps defensive pessimists to focus on developing good performance. Therefore, this theory suggests that defensive pessimists experience high levels of negative affect after negative reflection. In our study, a positive association was found between reflectivity scores and negative affect; a non-significant correlation was found between reflectivity scores and positive affect.

There is ample evidence that defensive pessimism helps anxious individuals to achieve successful performance in the short term: however, it is detrimental to emotional well-being in the long term [8]. Lei and Duan [36] suggested that Chinese college students' psychological health may be harmed by negative expectations. In a 3-year longitudinal study, students with high levels of defensive pessimism showed global life stress, psychological symptoms, and less satisfaction with their lives [37]. In relation to affect, Norem and Illingworth [10] suggested that although the induction of positive mood could improve the emotional well-being of defensive pessimists, it could worsen their performance.

This study has several limitations. Firstly, the results of the study are limited by its exclusive reliance on self-report measures. In addition, the cross-sectional study design means that causal associations cannot be identified and nature of the data leave open the possibility that directions of the associations could be different from those described. Longitudinal methods could be used in future studies to investigate the predictive value of defensive pessimism. Since strategic optimism, understood as a counterpoint to defensive pessimism, is the most widely used variable in research on this topic [38], this study is limited by the fact that this variable was not used to analyse the convergent validity of the DPQ-SV. Finally, the sample characteristic (i.e., undergraduate university students) might limit the generalizability of the current results.

In conclusion, the results suggest that the DPQ-SV has adequate stability, reliability, and criterion validity, and as such it appears to be a reliable tool for measuring defensive pessimism. Future research should use different samples in order to test the role of defensive pessimism not only in academic settings. As Seery et al. [9] suggested, if negative reflection leads to threat across situations for defensive pessimists, they could develop mental and physical health problems over time. Therefore, the study of defensive pessimism could be relevant not only in academic settings but also in healthcare settings.

## Supporting information

**S1 Table. Mardia tests of multivariate skew and kurtosis.**
(DOCX)

**S1 Fig. Multivariate outliers.** Model 1 (left).
(TIF)

**S2 Fig. Multivariate outliers.** Model 2 (right).
(TIF)

**S3 Fig. Multivariate outliers.** Reflectivity (Model 3).
(TIF)

**S4 Fig. Multivariate outliers.** Negative Expectation (Models 3 & 4).
(TIF)

**S5 Fig. Multivariate outliers.** Reflectivity (Model 4).
(TIF)

## Author Contributions

**Conceptualization:** Carmen Ramírez-Maestre, Rosa Esteve, Alicia E. López-Martínez.

**Data curation:** Carmen Ramírez-Maestre, Rosa Esteve, Elena R. Serrano-Ibáñez, Gema T. Ruiz-Párraga.

**Formal analysis:** Carmen Ramírez-Maestre, Rosa Esteve, Alicia E. López-Martínez.

**Funding acquisition:** Carmen Ramírez-Maestre, Rosa Esteve.

**Investigation:** Carmen Ramírez-Maestre, Rosa Esteve, Elena R. Serrano-Ibáñez, Alicia E. López-Martínez.

**Methodology:** Carmen Ramírez-Maestre, Teresa Rivas-Moya.

**Supervision:** Carmen Ramírez-Maestre, Rosa Esteve, Elena R. Serrano-Ibáñez, Alicia E. López-Martínez, Gema T. Ruiz-Párraga, Teresa Rivas-Moya.

**Validation:** Carmen Ramírez-Maestre, Teresa Rivas-Moya.

**Visualization:** Carmen Ramírez-Maestre.

**Writing – original draft:** Carmen Ramírez-Maestre.

**Writing – review & editing:** Carmen Ramírez-Maestre, Elena R. Serrano-Ibáñez, Alicia E. López-Martínez, Gema T. Ruiz-Párraga, Teresa Rivas-Moya.

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
