## [Decision Letter · Decision Letter 0]

16 Dec 2019

PONE-D-19-28221

Psychometric Characteristics and Factorial Structures of the Defensive Pessimism Questionnaire - Spanish Version (DPQ-SV).

PLOS ONE

Dear Dr. Ramírez-Maestre,

Thank you for submitting your manuscript to PLOS ONE. After careful consideration, we feel that it has merit but does not fully meet PLOS ONE’s publication criteria as it currently stands. Therefore, we invite you to submit a revised version of the manuscript that addresses the points raised during the review process.

We would appreciate receiving your revised manuscript by Jan 26 2020 11:59PM. To enhance the reproducibility of your results, we recommend that if applicable you deposit your laboratory protocols in protocols.io, where a protocol can be assigned its own identifier (DOI) such that it can be cited independently in the future. For instructions see: http://journals.plos.org/plosone/s/submission-guidelines#loc-laboratory-protocols

We look forward to receiving your revised manuscript.

Kind regards,

Francesca Chiesi

Academic Editor

PLOS ONE

Additional Editor Comments:

As minor comments, I suggest to use the subtitle "Construct vlidity" instead of "Convergent validity", and to replace the term Divergent with Discriminant.

Additionally, Cohen effect size cut-offs should be replaced with some specific validity criteria (e.g.,  EFPA Review Model for the Description and Evaluation of Psychological and Educational Tests, version 4.2.6. [(Available online: http://www.efpa.eu/download/650d0d4ecd407a51139ca44ee704fda4.

**Journal requirements:**

**When submitting your revision, we need you to address these additional requirements:**

**Please ensure that your manuscript meets PLOS ONE's style requirements, including those for file naming. The PLOS ONE style templates can be found at http://www.plosone.org/attachments/PLOSOne_formatting_sample_main_body.pdf and http://www.plosone.org/attachments/PLOSOne_formatting_sample_title_authors_affiliations.pdf**Thank you for including your dual publication statement.  To comply with PLOS ONE submission guidelines can you please upload a copy of the previous publication as a supplementary file.  Please be aware that if we locate a dual publication issue we will require copyright holder’s permission to publish under CC-By 4.0. We note that you have indicated that data from this study are available upon request. PLOS only allows data to be available upon request if there are legal or ethical restrictions on sharing data publicly. For more information on unacceptable data access restrictions, please see http://journals.plos.org/plosone/s/data-availability#loc-unacceptable-data-access-restrictions.In your revised cover letter, please address the following prompts:a) If there are ethical or legal restrictions on sharing a de-identified data set, please explain them in detail (e.g., data contain potentially sensitive information, data are owned by a third-party organization, etc.) and who has imposed them (e.g., an ethics committee). Please also provide contact information for a data access committee, ethics committee, or other institutional body to which data requests may be sent.b) If there are no restrictions, please upload the minimal anonymized data set necessary to replicate your study findings as either Supporting Information files or to a stable, public repository and provide us with the relevant URLs, DOIs, or accession numbers. For a list of acceptable repositories, please see http://journals.plos.org/plosone/s/data-availability#loc-recommended-repositories.We will update your Data Availability statement on your behalf to reflect the information you provide.

4. Please include your tables as part of your main manuscript and remove the individual files.

Please note that supplementary tables should be uploaded as separate "supporting information" files.

Reviewers' comments:

Reviewer's Responses to Questions

**Comments to the Author**

1. Is the manuscript technically sound, and do the data support the conclusions?

Reviewer #1: Yes

2. Has the statistical analysis been performed appropriately and rigorously? 

Reviewer #1: Yes

3. Have the authors made all data underlying the findings in their manuscript fully available?

Reviewer #1: Yes

4. Is the manuscript presented in an intelligible fashion and written in standard English?

Reviewer #1: Yes

5. Review Comments to the Author

Reviewer #1: I appreciate the opportunity PLOS ONE gives me to review this manuscript. In general terms, it is a relevant and methodologically well elaborated work. In my view, it is potentially publishable, although some modifications are need in order to definitely recommend its acceptance. These modifications are indicated below.

Introduction:

The introductory framework is clear in terms of its structure and justification of the objective of the study. However, some aspects should be delved deeper:

- In my opinion, the explanation about the conceptual and empirical distinction between negative expectations and reflection is very superficial. At a conceptual level, it is necessary to clarify in greater detail the characteristics of both constructs. From an empirical point of view, the lack of consensus in research must be deepened as to whether defensive pessimism should be considered a one-dimensional or two-dimensional construct and, even, if negative expectations and reflection should be considered independent strategies (see, for example, Gasper, Lozinski, & LeBeau, 2009; Martin, Marsh, & Debus, 2001a, 2001b, 2003*).

This issue can also contribute to enriching the Discussion.

- Lines 111-112: It should be conveniently justified why the authors hypothesize to find a positive correlation between the two DPQ-SV subscales and trait anxiety and negative affect, as well as a negative correlation with positive affect. In this sense, perhaps it would be convenient for the authors to deepen the consequences of using the strategy of defensive pessimism, in the short and long term.

- Lines 113-114: “Finally, we assessed the role of defensive pessimism dimensions as a mediator between anxiety and affect”. It should be justified why this mediation analysis contributes to the validation of the scale and, if so, what hypotheses the authors raise regarding this analysis.

- Lines 86-87: “Defensive pessimism is a cognitive strategy that individuals with high levels of anxiety use to have success in their performance”. I think this phrase is not correct, and does not reflect the essence of defensive pessimism. According to the phrase, it seems that defensive pessimists use their strategy with an intent to achieve success, which is not the case. Defensive pessimism moves between the orientation to success and the fear of failure, that is, cognitively they are committed to fear, but behaviorally they move between the approach to success and the avoidance of failure (Martin and Marsh, 2003). Therefore, I recommend that the authors rectify the phrase contained in lines 86-87.

- Line 111: What is lot? These acronyms are clarified in the abstract, but their meaning should also be indicated in the body of the manuscript.

Participants:

- How was the sample obtained? What sampling procedure was followed?

- It is indicated that the sample producing data was 539 subjects, but the abstract indicates that it was 542. What is the real number?

Instruments:

- DPQ: the reliability of this instrument should be indicated in the Results section, not here.

- LOT-R: was any version validated in the Spanish context used? In the rest of the instruments its Spanish version is referenced, but not in the LOT-R.

Results:

Regarding the convergent validity, and in view of the existence of two factors, it would be advisable to analyze the average variance extracted.

Discussion:

In general, this section requires more elaboration.

- For example, it would be necessary for the authors to provide a plausible explanation about the non-significant relationship between reflection and positive affect (as well as the moderately positive relationship with negative affect). One possibility would be to analyze this issue in the light of the hypothesis of dissipation and harnessing of threat (e.g., Seery, West, Weisbuch, & Blascovich, 2008).

- Lines 339-343: In my opinion, the conceptual justification provided regarding the elimination of item 10 is not correct. The authors argue that defensive pessimists do not consider positive results. This does not seem to be the case, at least as regards the reflection component. According to Norem (2002, p. 111), one of the positive characteristics of defensive pessimists is hope, a psychological quality that makes them hold expectations of control over their own actions and, therefore, to find a way to achieve what they want. From this consideration, the authors should justify in a clearer and unambiguous way, why they consider that item 10 does not reflect the essence of defensive pessimism.

- The limitations of the study should be indicated. From my point of view, among these limitations would be the fact of not having used the strategic optimism variable to analyze the convergent validity of the DPQ-SV. Strategic optimism is the most widely used variable in research as a counterpoint to defensive pessimism (see Norem, 2008).

References:

The reference number 11 is not cited within the manuscript.

* Gasper, K., Lozinski, R., & LeBeau, L. S. (2009). If you plan, then you can: How reflection helps defensive pessimists pursue their goals. Motivation and Emotion, 33(2), 203-216. doi: 10.1007/s11031-009-9125-5

Martin, A. J., & Marsh, H. W. (2003) Fear of failure: friend or foe? Australian Psychologist, 38(1), 31-38. doi: 10.1080/00050060310001706997

Martin, A. J., Marsh, H. W., & Debus, R. L. (2001a). A quadripolar need achievement representation of self-handicapping and defensive pessimism. American Educational Research Journal, 38(3), 583-610. doi: 10.1037/0022-0663.95.3.617

Martin, A. J., Marsh, H. W., & Debus, R. L. (2001b). Self-handicapping and defensive pessimism: Exploring a model of predictors and outcomes from a self-protection perspective. Journal of Educational Psychology, 93(1), 87-102. doi: 10.1037/0022-0663.93.1.87

Martin, A. J., Marsh, H. W., & Debus, R. L. (2003). Self-handicapping and defensive pessimism: A model of self-protection from a longitudinal perspective. Contemporary Educational Psychology, 28, 1-36. doi: 10.1016/s0361-476X(02)00008-5

Norem, J. K. (2002). The positive power of negative thinking. Basic Books: New York.

Norem, J. K. (2008). Defensive pessimism, anxiety, and the complexity of evaluating self-regulation. Social and Personality Psychology Compass, 2, 121-134. doi: 10.1111/j.1751.9004.2007.00053x

Seery, M., West, T., Weisbuch, M., & Blascovich, J. (2008). The effects of negative reflection for defensive: Dissipation or harnessing of threat? Personality and Individual Differences, 45(6), 515-520. doi: 10.1016/j.paid.2008.06.004

6. PLOS authors have the option to publish the peer review history of their article (what does this mean?). If published, this will include your full peer review and any attached files.

Reviewer #1: No

---

## [Author Response · Author response to Decision Letter 0]

22 Jan 2020

Málaga, January 2019

PLOS ONE

 Dear Dr. Chiesi,

We very much appreciate your new suggestions regarding our paper (PONE-D-19-28221

Psychometric Characteristics and Factorial Structures of the Defensive Pessimism Questionnaire - Spanish Version (DPQ-SV). We have revised the paper accordingly and hope the new version is now acceptable for publication.

In the following we explain how the reviewers' suggestions have been taken into account in the new version. Changes in the text can be found in red.

Firstly, we have realised that, since the correlation between reflectivity (mediator) and positive affect (criterion variable) was non-significant, according to Baron and Kenny [33], the mediator role of reflectivity between trait-anxiety and positive affect could not be calculated. Therefore, we have conducted the mediation analysis again. Although the new results are very similar to the previous ones, several differences can be seen in Table 5.

Editor’s comments:

As minor comments, I suggest to use the subtitle "Construct validity" instead of "Convergent validity", and to replace the term Divergent with Discriminant.

Answer. Since the Factorial Analysis has previously been described in the text and provides evidence of construct validity, the subtitle ‘Other Evidence of Construct Validity’ has been included. This section now includes convergent validity.

Additionally, Cohen effect size cut-offs should be replaced with some specific validity criteria (e.g., EFPA Review Model for the Description and Evaluation of Psychological and Educational Tests, version 4.2.6. [(Available online: http://www.efpa.eu/download/650d0d4ecd407a51139ca44ee704fda4.

Answer. Specific criteria values given by Evers et al. (2013) have now been used in convergent and criterion validity. This reference has also been included.

Reviewer comments:

Reviewer #1 (R#1)

I appreciate the opportunity PLOS ONE gives me to review this manuscript. In general terms, it is a relevant and methodologically well elaborated work. In my view, it is potentially publishable, although some modifications are need in order to definitely recommend its acceptance. These modifications are indicated below.

Answer. Thank you very much for your positive comments.

Introduction: The introductory framework is clear in terms of its structure and justification of the objective of the study. However, some aspects should be delved deeper:

- In my opinion, the explanation about the conceptual and empirical distinction between negative expectations and reflection is very superficial. At a conceptual level, it is necessary to clarify in greater detail the characteristics of both constructs. From an empirical point of view, the lack of consensus in research must be deepened as to whether defensive pessimism should be considered a one-dimensional or two-dimensional construct and, even, if negative expectations and reflection should be considered independent strategies (see, for example, Gasper, Lozinski, & LeBeau, 2009; Martin, Marsh, & Debus, 2001a, 2001b, 2003*).

This issue can also contribute to enriching the Discussion.

Answer. Thank you for your comments. We have included more information on the nature of the relationship between negative expectations and reflection (see red text in the introduction and discussion sections): 

(lines 83-99) Individuals with high scores in defensive pessimism set unrealistically low expectations for upcoming performance, even though they have had previous successes, and then devote considerable energy to mentally playing through or reflecting on all the possible outcomes they can imagine for a given situation. Thus, defensive pessimism includes two different domains: negative expectations about success in a particular task (pessimism) and reflection [6]. These two variables may act independently in the behavior of defensive pessimists [7]. The role of reflection is complex and has two different explanations: the dissipation hypothesis and the harnessing hypothesis [8]. The dissipation hypothesis suggests that when defensive pessimists reflect about worst-case scenarios, they dissipate negative affect before performance. However, according to the harnessing hypothesis, when they reflect about potential negative outcomes regarding an upcoming task there is an increase in negative affect (anxiety), which helps defensive pessimists to focus on developing good performance [7; 9]. More empirical research is needed to support one of these two theoretical explanations.

(lines 108-113) In the 1980s, Norem and Cantor [5] developed a unidimensional questionnaire (negative expectations) to assess defensive pessimism. Subsequently, the Defensive Pessimism Questionnaire (DPQ) included two correlated factors called Reflectivity and Pessimism that assessed both domains: negative expectations and reflection [8]. Although the DPQ has been viewed as a unidimensional measure of defensive pessimism [12], factorial analysis shows that negative expectations and reflection are different but correlated factors [13]. 

(lines 392-398) Previous research has shown that individuals high in pessimism (both defensive and typical) are characterized by negative expectations about the future: however, whereas typical pessimists are passive, defensive pessimists reflect and plan how to manage future events [8]. The union of both Negative Expectations and Reflectivity differentiate defensive and typical pessimism. Thus, although the DPQ has been viewed as a unidimensional measure of defensive pessimism [12], factorial analysis has found two correlated factors [6, 8, 13]. Our results support this finding

- Lines 111-112: It should be conveniently justified why the authors hypothesize to find a positive correlation between the two DPQ-SV subscales and trait anxiety and negative affect, as well as a negative correlation with positive affect. In this sense, perhaps it would be convenient for the authors to deepen the consequences of using the strategy of defensive pessimism, in the short and long term.

Answer. Thank you for your suggestions. We agree with you and, therefore, new text has been included in the introduction section (new text shown in red), explaining that (lines 121-136) According to theoretical considerations [4, 10], a moderate-high association was expected between negative expectation scores, LOT-pessimism scores (positive) and LOT-optimism scores (negative), and a low-moderate correlation between reflectivity scores, LOT-pessimism scores (positive) and LOT-optimism scores (negative). In order to test concurrent criterion validity, we measured the association between defensive pessimism and affect. According to the harnessing hypothesis [8], negative expectations and reflection increase negative affect, which helps defensive pessimists to obtain better performance outcomes. Therefore, we expected to find moderate-strong positive correlations between DPQ-SV subscale scores and negative affect, and moderate-strong negative correlations between DPQ-SV subscale scores and positive affect. Finally, it has been suggested that defensive pessimists are typically high in trait-anxiety [4, 5, 14]. It has also been found that defensive pessimism, understood as a cognitive strategy that anxious individuals use in the face of challenging situations, increases negative affect and decreases positive affect [4,6,10]. Therefore, in order to test the validity of the DPQ-SV, we also assessed the mediating role of the dimensions of defensive pessimism between trait-anxiety and affect.

Regarding the consequences of using the strategy of defensive pessimism, the following new text has been included: (lines 428-436) There is ample evidence that defensive pessimism helps anxious individuals to achieve successful performance in the short term: however, it is detrimental to emotional well-being in the long term [8]. Lei and Duan [35] suggested that Chinese college students’ psychological health may be harmed by negative expectations. In a 3-year longitudinal study, students with high levels of defensive pessimism showed global life stress, psychological symptoms, and less satisfaction with their lives [36]. In relation to affect, Norem and Illingworth [10] suggested that although the induction of positive mood could improve the emotional well-being of defensive pessimists, it could worsen their performance. 

- Lines 113-114: “Finally, we assessed the role of defensive pessimism dimensions as a mediator between anxiety and affect”. It should be justified why this mediation analysis contributes to the validation of the scale and, if so, what hypotheses the authors raise regarding this analysis.

Answer. In line with these comments, the following sentence has been included in the introduction section: “(lines 131-136) Finally, it has been suggested that defensive pessimists are typically high in trait-anxiety [4, 5, 14]. It has also been found that defensive pessimism, understood as a cognitive strategy that anxious individuals use in the face of challenging situations, increases negative affect and decreases positive affect [4,6,10]. Therefore, in order to test the validity of the DPQ-SV, we also assessed the mediating role of the dimensions of defensive pessimism between trait-anxiety and affect.” And in discussion section: (lines 406-411) We also found that negative expectations and reflectivity factors mediated between trait-anxiety and negative affect. These results support those of Araújo et al. [34], who analysed the relationship between defensive pessimism and psychological health in a sample of 192 university students. These authors found that the low expectations set by defensive pessimists fully mediated the relationship between anxiety and poor psychological health

- Lines 86-87: “Defensive pessimism is a cognitive strategy that individuals with high levels of anxiety use to have success in their performance”. I think this phrase is not correct, and does not reflect the essence of defensive pessimism. According to the phrase, it seems that defensive pessimists use their strategy with an intent to achieve success, which is not the case. Defensive pessimism moves between the orientation to success and the fear of failure, that is, cognitively they are committed to fear, but behaviorally they move between the approach to success and the avoidance of failure (Martin and Marsh, 2003). Therefore, I recommend that the authors rectify the phrase contained in lines 86-87.

Answer 5: Thank you for your comment. In line with this suggestion, this sentence has been deleted. It now reads: (lines 86-89) Individuals with high scores in defensive pessimism set unrealistically low expectations for upcoming performance, even though they have had previous successes, and then devote considerable energy to mentally playing through or reflecting on all the possible outcomes they can imagine for a given situation

- Line 111: What is lot? These acronyms are clarified in the abstract, but their meaning should also be indicated in the body of the manuscript.

Answer: We agree with the reviewer. It now reads: the Life Orientation Test (LOT). Thank you for the suggestion.

Participants: - How was the sample obtained? What sampling procedure was followed?- It is indicated that the sample producing data was 539 subjects, but the abstract indicates that it was 542. What is the real number?

Answer: We are sorry for this mistake. The sample included 539 students. The text now reads: (lines 140-147) During the course 2017/18, random cluster sampling was conducted among the 1080 undergraduate students of the Faculty of Psychology of Malaga University. Firstly, two groups from each course were randomly selected. The questionnaires were administered to all students in each group. The sample size was 551 students from the four courses (199, 139, 113, 100 students from each course, respectively). Twelve participants had missing scores and so the final sample size comprised 539 students. The inclusion criterion was the ability to understand the Spanish language. Women comprised 78.8% of the sample. Mean age was 21.18 years (SD = 4.53; range 18–53)

Instruments: 

- DPQ: the reliability of this instrument should be indicated in the Results section, not here.

- LOT-R: was any version validated in the Spanish context used? In the rest of the instruments its Spanish version is referenced, but not in the LOT-R.

Answer: Following reviewer's 1 suggestion, the reliability of the DPQ has been deleted from Instrument section. The Spanish version of LOT-R has also been cited. Thank you. 

Results: Regarding the convergent validity, and in view of the existence of two factors, it would be advisable to analyze the average variance extracted.

Answer: Following reviewer's 1 suggestion, the average variance percentage explained by the two components is shown in the text. Therefore, both DPQ-SV subscales (Negative Expectation and Reflectivity) showed an adequate convergence with LOT-optimism and LOT-pessimism. In addition, the two components of defensive pessimism explained a greater average variance percentage of the LOT-pessimism scores (24%) than of the LOT-optimism scores (18.33%).

Discussion: In general, this section requires more elaboration.

- For example, it would be necessary for the authors to provide a plausible explanation about the non-significant relationship between reflection and positive affect (as well as the moderately positive relationship with negative affect). One possibility would be to analyze this issue in the light of the hypothesis of dissipation and harnessing of threat (e.g., Seery, West, Weisbuch, & Blascovich, 2008).

Answer: Thank you very much. This has been a very useful suggestion. We have included new text in the discussion section, explaining the results in the light of the harnessing hypothesis. 

(lines 406-427) We also found that negative expectations and reflectivity factors mediated between trait-anxiety and negative affect. These results support those of Araújo et al. [34], who analysed the relationship between defensive pessimism and psychological health in a sample of 192 university students. These authors found that the low expectations set by defensive pessimists fully mediated the relationship between anxiety and poor psychological health. On the other hand, we have found that both Negative Expectations and Reflectivity had a positive (high and moderate respectively) correlation with negative affect. Interestingly, the correlation between reflectivity and positive affect was not significant. In line with previous studies, these results support the harnessing hypothesis versus the dissipation hypothesis [7, 9]. According to the dissipation hypothesis, when defensive pessimists reflect about worst-case scenarios, they dissipate negative affects before performance. In relation to this theoretical framework, Norem & Chang [8] suggested that individuals with high levels of anxiety could use defensive pessimism as a way to cope with their negative affect to achieve successful performance. Norem & Cantor [4] also proposed that defensive pessimists control their anxiety by thinking through possible outcomes prior to an event. However, according to the harnessing hypothesis, when defensive pessimists reflect on potential negative outcomes for an upcoming task there is an increase in negative affect, which helps defensive pessimists to focus on developing good performance. Therefore, this theory suggests that defensive pessimists experience high levels of negative affect after negative reflection. In our study, a positive association was found between reflectivity scores and negative affect; a non-significant correlation was found between reflectivity scores and positive affect.

- Lines 339-343: In my opinion, the conceptual justification provided regarding the elimination of item 10 is not correct. The authors argue that defensive pessimists do not consider positive results. This does not seem to be the case, at least as regards the reflection component. According to Norem (2002, p. 111), one of the positive characteristics of defensive pessimists is hope, a psychological quality that makes them hold expectations of control over their own actions and, therefore, to find a way to achieve what they want. From this consideration, the authors should justify in a clearer and unambiguous way, why they consider that item 10 does not reflect the essence of defensive pessimism.

Answer. Thank you very much for your interesting reflections. In fact, we did consider item 10, but the results of the Robust Confirmatory Factor Analysis (RCFA) showed that model 4 (two factors without item 10) had the best fit. Although Norem (2001) stated that people who tend to reflect extensively about possible negative outcomes also tend to reflect extensively about possible positive outcomes (and therefore, item 10 should be part of reflexivity factor), our results (similar to those of Lim) show that the quality of the instrument could be improved without this item. Lim’s explanation was that defensive pessimists do not consider positive results. We have changed the text in order to clarify this. 

(lines 381-398) Lim [13] suggested that item 10 may not adequately capture the essence of the defensive pessimism construct, given that defensive pessimists are concerned about the possibility of negative events but do not consider positive outcomes. However, Norem [8] argued that people who tend to reflect extensively about possible negative outcomes also tend to reflect extensively about possible positive outcomes, and that both thoughts are part of the reflection process of defensive pessimists. In any case, according to our results, the quality of the DPQ could be increased without item 10. Our results, and those of Lim [13], could simply be the consequence of the existence of only one item assessing reflection about positive outcomes; if more items regarding positive outcomes had been included, the results of the factor analysis may not have indicated the exclusion of item 10. 

Previous research has shown that individuals high in pessimism (both defensive and typical) are characterized by negative expectations about the future: however, whereas typical pessimists are passive, defensive pessimists reflect and plan how to manage future events [8]. The union of both Negative Expectations and Reflectivity differentiate defensive and typical pessimism. Thus, although the DPQ has been viewed as a unidimensional measure of defensive pessimism [12], factorial analysis has found two correlated factors [6, 8, 13]. Our results support this finding.

- The limitations of the study should be indicated. From my point of view, among these limitations would be the fact of not having used the strategic optimism variable to analyze the convergent validity of the DPQ-SV. Strategic optimism is the most widely used variable in research as a counterpoint to defensive pessimism (see Norem, 2008).

Answer: Following these suggestions, more limitations have been include in the discussion section (see text in red)

Since strategic optimism, understood as a counterpoint to defensive pessimism, is the most widely used variable in research on this topic [37], this study is limited by the fact that this variable was not used to analyse the convergent validity of the DPQ-SV. 

References: The reference number 11 is not cited within the manuscript.

Answer: We apologise for this mistake. It has been included along with several new references, thus, the references have been updated.

---

## [Editor Report · Decision Letter 1]

31 Jan 2020

PONE-D-19-28221R1

Psychometric Characteristics and Factorial Structures of the Defensive Pessimism Questionnaire - Spanish Version (DPQ-SV).

PLOS ONE

Dear Dr. Ramírez-Maestre,

Thank you for submitting your manuscript to PLOS ONE. After careful consideration, we feel that it has merit but does not fully meet PLOS ONE’s publication criteria as it currently stands. Therefore, we invite you to submit a revised version of the manuscript that addresses the points raised during the review process.

Thank you for submitting the revised version of your paper to PlosOne. 

I have read through your revised paper carefully and I consider that the comments have been adequately addressed. I have just two suggestions:

1. I suggest to present Table 3 and Figure 1 and 2 as Supporting Information/Material

2. I'd include in the limitations that the sample characteristic (i.e., undergraduate university students) might limit the generalizability of the current results. 

We would appreciate receiving your revised manuscript by Mar 16 2020 11:59PM. To enhance the reproducibility of your results, we recommend that if applicable you deposit your laboratory protocols in protocols.io, where a protocol can be assigned its own identifier (DOI) such that it can be cited independently in the future. For instructions see: http://journals.plos.org/plosone/s/submission-guidelines#loc-laboratory-protocols

We look forward to receiving your revised manuscript.

Kind regards,

Francesca Chiesi

Academic Editor

PLOS ONE

---

## [Author Response · Author response to Decision Letter 1]

3 Feb 2020

Málaga, January 2019

PLOS ONE

 Dear Dr. Chiesi,

We very much appreciate your new suggestions regarding our paper (PONE-D-19-28221

Psychometric Characteristics and Factorial Structures of the Defensive Pessimism Questionnaire - Spanish Version (DPQ-SV). We have revised the paper accordingly and hope the new version is now acceptable for publication.

Editor’s comments:

1. I suggest to present Table 3 and Figure 1 and 2 as Supporting Information/Material

2. I'd include in the limitations that the sample characteristic (i.e., undergraduate university students) might limit the generalizability of the current results. 

Answer. 

1. We have included Table 3 and Figure 1 and 2 as Supporting Information/Material

2. We have include in the limitations that the sample characteristic (i.e., undergraduate university students) might limit the generalizability of the current results.

---

## [Editor Report · Decision Letter 2]

12 Feb 2020

Psychometric Characteristics and Factorial Structures of the Defensive Pessimism Questionnaire - Spanish Version (DPQ-SV).

PONE-D-19-28221R2

Dear Dr. Ramírez-Maestre,

We are pleased to inform you that your manuscript has been judged scientifically suitable for publication and will be formally accepted for publication once it complies with all outstanding technical requirements.

With kind regards,

Francesca Chiesi

Academic Editor

PLOS ONE

---

## [Editor Report · Acceptance letter]

26 Mar 2020

PONE-D-19-28221R2 

Psychometric Characteristics and Factorial Structures of the Defensive Pessimism Questionnaire - Spanish Version (DPQ-SV). 

Dear Dr. Ramírez-Maestre:

I am pleased to inform you that your manuscript has been deemed suitable for publication in PLOS ONE. Congratulations! Your manuscript is now with our production department. 

With kind regards,

on behalf of

Dr. Francesca Chiesi 

Academic Editor

PLOS ONE